# Experimental Study on a Novel Shear Connection System for FRP-Concrete Hybrid Bridge Girder

**DOI:** 10.3390/ma13092045

**Published:** 2020-04-27

**Authors:** Mateusz Rajchel, Maciej Kulpa, Tomasz Siwowski

**Affiliations:** Department of Roads & Bridges, Rzeszow University of Technology, 35-959 Rzeszow, Poland; mrajchel@prz.edu.pl (M.R.); kulpa@prz.edu.pl (M.K.)

**Keywords:** bridge, FRP beam, LWC slab, hybrid girder, shear connection system, push-out test, fatigue test, Eurocode 4

## Abstract

The study presents experimental results of an investigation on a novel shear connection system for hybrid bridge girders composed of laminated composite beams and concrete slabs. The special connector comprised of a steel plate and welded bolts is attached to beam’s top flange by adhesive bonding and with a preset torque of nuts. The study’s purpose is to check ductility, safety, reliability and robustness of the shear connection before its implementation in the first Polish composite bridge. Three static push-out tests and fatigue test were performed to evaluate the shear connection behavior under static and cyclic loading. The load–slip curves, shear capacity, fatigue strength and failure mechanisms of the novel shear connectors are discussed. The high-slip modulus indicates that the connectors can very efficiently promote the composite action. The ultimate resistance and the fatigue strength obtained from the test was about 12% and 66% higher than the characteristic resistance and the fatigue strength of common headed studs, according to Eurocode 4, respectively. An estimated global safety factor of 3.67 showed the high safety, reliability and robustness of the novel connection system. The study discusses the structural performance of the proposed connection system, demonstrating its technical suitability.

## 1. Introduction

Since the beginning of the XXI century, fiber-reinforced polymer (FRP) composites have become an integral part of the construction industry because of their versatility, high strength-to-weight ratio, enhanced durability, resistance to fatigue and corrosion, accelerated construction as well as lower maintenance and life-cycle costs. Due to these excellent performance characteristics FRP composites offer very promising options for a wide range of civil infrastructure applications, particularly in bridge engineering [1,2,3,4,5,6,7,8]. However, FRP composites in bridge application are also faced with several challenges [9]. These include relatively low stiffness, existence of brittle failure modes and high initial cost. To address these challenges hybrid FRP–concrete bridges have been designed and constructed since the end of XX century [10,11,12,13,14]. The motivation for the design approach is to fully utilize the respective material properties of FRP and concrete, i.e., concrete’s stiffness and high compressive strength and FRP’s high tensile strength. In fact, there are several advantages with the FRP–concrete hybrid girders, referring to the increase of flexural stiffness thus reducing deformability and preventing the buckling phenomena of the FRP profiles or shells. Moreover, the “pseudo-ductile” behavior, necessary for bridges and the existence of composite action between the FRP beam and the concrete slab are a serious advantage of these girders [15,16,17,18,19,20].

In these hybrid systems, a strong FRP–concrete connection is an indispensable prerequisite to achieve structural integrity. The design of efficient slab-to-beam connection is the most challenging topic in the development of hybrid bridge girders. The efficiency of slab-to-beam joint governs the overall behavior of the formed bridge superstructure. Composite action, structural redundancy and system ductile characteristic are main objectives when designing connections for hybrid girders. In connections with composite action, the efficiency of shear transfer and constructability are major factors directing the connection design. The need for efficient and reliable load carrying joints that can endure long-term fatigue and environmental attacks has become apparent.

Concluding from the existing literatures, several types of shear connections have been researched, listed as follow: coated sand layer [21], adhesive [22], ribs and indents [23,24], steel studs or bolts [19,25,26], FRP dowels [27], FRP shear key [28] and perforated FRP rib [29]. Hence, far, most FRP–concrete connections have utilized joining in which conventional steel shear studs or bolts are used to provide composite action. These types of connections have a proven history in the civil engineering domain, therefore are generally accepted by bridge engineers. However, these connections are developed for common steel–concrete composite bridge girders. Since FRP beams are different from steel beams in constitutive materials and structural forms, efforts shall be taken to develop adapted connection techniques for hybrid FRP–concrete girders. The steel shear studs have relatively high shear capacity, but large slippage occurred along the interface, i.e., a lower composite action was obtained. Moreover, in shear stud connections holes are often required at some desired spacing in the top flanges of the FRP beam. The cut-outs bring concerns about the stress concentrations and possible vulnerability to fatigue loads and environmental attacks in the FRP beams cut-out regions. Efforts must be taken to protect the cut-out regions from environmental attacks. Although several connections which resemble steel shear studs have been proposed and tested, the lack of test data and design recommendations limits the use of FRP–concrete hybrid girders. To address the above mentioned concerns the modified connection system comprising welded bolts and epoxy adhesive was developed and implemented in the first Polish FRP composite bridge.

The innovative hybrid idea of a FRP composite–lightweight concrete structural system for the heaviest traffic load class bridges has been proposed and accepted as the subject of the demonstrative R&D project ComBridge, performed by the research-industry consortium led by Mostostal Warszawa S.A., one of the biggest Polish contractor in the field of civil infrastructure. As a result, the first Polish FRP bridge was developed and built in late autumn 2015 [14]. Due to the lack of commonly accepted design procedures, the whole designing process was strongly supported by laboratory testing. The comprehensive material testing and the structural tests were carried out, comprising among others: the evaluation of strength and stiffness of a lightweight concrete (LWC) deck slab reinforced with glass fiber-reinforced polymer (GFRP) composite bars [30], the strength and fatigue tests on connections between LWC slab and FRP beam (reported in this study), the comprehensive research on the structural behavior of the full-scale hybrid FRP composite–concrete bridge girder [20] and finally the field evaluation of the hybrid FRP–concrete road bridge [31].

This study presents the results of experimental research concerning the shear behavior of the novel connection system between the FRP beam and the concrete slab. The special connector comprised of a steel plate and welded bolts is attached to beam’s top flange by adhesive bonding and with a preset torque of nuts. Several push-out tests were performed on connector’s specimens made of composite laminates connected to concrete slabs. The connection performance is studied with particular relevance to slip behavior and shear capacity. Further analysis reveals the failure mechanism and the fatigue strength of the connection system as well. The study summarizes the results of the static and fatigue tests and discusses the structural performance of the proposed connection system, demonstrating its technical suitability. Furthermore, the estimated global safety factor shows the high safety, reliability and robustness of the novel connection system. The applicability of Eurocode 4 [32] procedures to predict static and fatigue strength of the connection and to check its ULS/SLS design provisions is also demonstrated.

## 2. FRP-Concrete Hybrid Bridge Girder and Shear Connection System

The novel hybrid FRP-concrete bridge girder was designed to be used in the first Polish road bridge made of FRP composites [14]. The hybrid girder consists of two main parts: a FRP composite beam with open, trapezoidal cross-section and a concrete slab (Figure 1). The FRP composite beam is 1025 mm deep, 1550 mm wide at the top and 621 mm wide at the bottom. The both top flanges of beam are 301 mm wide and are integrated with the bottom plate by means of two slightly inclined webs. The individual laminates have the thickness of about 25 mm, 22 mm and 19 mm for top flanges, webs and bottom plate, respectively. The top flanges and webs are made of solid and sandwich GFRP laminates, respectively, whereas the bottom plate has a hybrid glass/carbon fiber-reinforced polymer solid structure [20]. On two top flanges the stay-in-place (SIP) formwork is supported to facilitate concrete slab casting. The SIP formwork is made of a 33.4 mm thick sandwich plate built alike webs.

The 180 mm thick slab was made of 35/38 class LWC based on the lytag type aggregate with density ρ = 1968 kg/m^3^. The slab was reinforced longitudinally and transversally with two grids made of 12 mm ribbed GFRP rebars spaced every 120 mm in each direction. The LWC slab was connected to FRP beam by means of novel shear connectors made of M20 class 4.8 bolts welded to a rectangular steel plate (Figure 2). The single connector consisted of 8 bolts welded in two rows to 10 mm thick steel plate of 240 × 660 mm in plan. The connectors were attached to bottom surface of beam’s top flanges with epoxy adhesive.

The developed shear connection system combines two convectional means used for slab-to-beam connections in hybrid FRP-concrete girders: studs and adhesive [29]. For studs the typical M20 bolts are used with belonging nuts and washers. Several bolts are welded to a steel plate in one or two rows with transverse and longitudinal spacing according to an appropriate design (Figure 2). The connector is deep galvanized to achieve protection against corrosion compared to FRP composite. To fasten the connector to the beam’s top flange, holes drilling is required in laminate, and the number, arrangement and diameter of holes correspond to design layout of studs. Subsequently the adhesive layer is place on the relevant areas from the bottom of the girder’s top flange and the connector is attached to laminate by bonding. Finally, steel washer and two nuts are installed on each bolt: the first screwed with a preset torque helps a proper bonding, the latter is placed on the top of the bolt to create a headed stud and to increase tear-out capacity of the connection (Figure 2).

The novel connection system differs from common studs in steel–concrete composite systems and bolts in FRP-concrete hybrid systems. In steel-concrete composite systems the headed studs are usually welded to top flange of steel girder. In FRP-concrete hybrid systems shear connectors are usually bolted to the FRP top flange using a preset torque. Here, combining welded headed bolts with a preset torque and adhesive in the connection a relatively high shear capacity is ensured and slippage along the interface is expected to be considerably limited. Moreover, epoxy placed around the laminate holes and at the interface protects the cut-out regions from environmental attacks and stress release is also induced in the laminate around the hole.

## 3. Shear Connection Design

Despite the load-slip behavior of the shear FRP-concrete hybrid connections is different from that in steel-concrete composite systems [19,25,26], the conventional design method provided in Eurocode 4 [32] was used in bridge design. It was assumed, that the welded bolts transmitted the longitudinal shear force between the concrete and the FRP beam, ignoring the effect of natural bond between the two and neglecting adhesive bond as well. The push-out testing reported in this study was undertaken to check if such an assumption were valid for the novel connection system, which could facilitate its further design. Thus, the applicability of Eurocode 4 [32] procedures to predict static and fatigue strength of the connection and to check its ULS/SLS design provisions was also examined.

The design procedure (described in Appendix A) based on Eurocode 4 [32] provisions was applied to determine the exact number and spacing of shear studs in the hybrid girder taking into account the size and strength of M20 bolts chosen for studs. As a result, along the girder with the length of 21.0 m the total number of 480 studs were mounted in 4 rows (two rows per a flange) with longitudinal spacing of 180 mm. For such a stud arrangement the relevant shear forces per unit length v_L,Ed_ and corresponding shear forces V_Ed_ were determined according to Formulas (A6)–(A8) in Appendix A and thus the shear force per one stud was established to be assumed in testing program (Table 1).

## 4. Static Tests

To evaluate the strength of shear connection system the push-out tests were carried out on full-scale specimens of shear connectors. Since there is no standard for testing this type of joints with FRP elements, it was decided to use the specimen presented in the main part of Eurocode 4 [33] for joints validation in steel–concrete composite structures. Specific push-out tests were carried out and the slabs and the reinforcement were suitably dimensioned in comparison with the girders for which the test was designed. The bonding at the interface between GFRP flanges of the beam and the concrete slab was not prevented as did actually on-site. From these push-out tests the load-slip and load-displacement performance, the failure load and the mode of failure were obtained.

### 4.1. Specimens’ Fabrication

The specimens for push-out tests were composed of two GFRP C-sections, bonded together to create a rectangular tube, and two concrete slabs joined to GFRP by shear connectors (Figure 3). Fabrication of specimens included: (a) manufacturing GFRP sections and bonding them in the shape of rectangular tube, (b) drilling holes in relevant laminates of the tube with predefined spacing and diameter, (c) fabricating two steel connectors by welding bolts to steel plates, (d) setting and bonding the steel connectors inside the tube with epoxy adhesive, (e) making the plywood formwork and setting the embedded GFRP reinforcing bars; (f) casting and curing the concrete (Figure 4). The total of 6 specimens were prepared: three for static tests (denoted S1–S3) and the subsequent three for fatigue test (denoted F4–F6).

The GFRP C-type sections were fabricated by infusion and their laminates had the same thickness and stacking sequence as the top flange of the girder. Structural epoxy adhesive was used to join both sections in a rectangular tube with the outer dimensions of 535 × 390 × 303 mm. The steel connectors were made of two M20 class 4.8 normal strength bolts spaced at 100 mm and welded to S355 class steel plate of 250 × 240 × 10 mm. The connectors were attached by bonding to the GFRP tube with structural epoxy adhesive SPABOND 340 LV HT with 3 mm bondline. The bolts were embedded in the concrete slab on the length of 150 mm. The relevant steel washers were used to disperse the stress caused by torqueing nuts. Additional nuts were screwed at bolts’ top to serve as a head. Two 390 × 375 × 180 mm concrete slabs were made of the LWC class 35/38 reinforced with two grids (per slab) made of 12 mm ribbed GFRP bars with spacing of 80 × 80 mm.

The basic material properties of GFRP’s laminas and LWC used for specimens’ elements were determined by testing and are listed in Table 2 and Table 3. The basic material properties of steel bolts class 4.8 were taken from the relevant code [34] as follows: tensile strength f_u_ = 400 MPa, yield strength f_y_ = 320 MPa and modulus of elasticity E = 200 GPa.

### 4.2. Test Setup

The specimens were tested in the universal test machine Schenck with 630-kN capacity. During the static tests, the rotation of the two lateral LWC blocks was prevented. Load was applied to the specimens by a hydraulic actuator. To prevent local failure of laminates the load was applied to the specimens through a set of steel plates placed on the laminates by epoxy layer (Figure 5). The loading sequence of each specimen was as follows: five cycles of loading up to characteristic load level of P_Ed,k_ = 87.92 kN, subsequently five cycles of loading up to design load level of P_Ed,d_ = 125.36 kN, and finally loading up to specimen’s failure. The both intermediate load levels were determined taking into account the maximum shear force per stud (Table 1) multiplied by four studs of the specimen. The loading rate was 2.0 kN/s.

Four linear variable differential transducers (LVDTs) were installed to measure the relative slip between the GFRP and the concrete. Figure 6 shows the location of these LVDTs on the specimen (measurement points P1/1, P1/2, P3/1 and P3/2). Additionally, two LVDTs were installed to check the loading conditions and displacement of the specimen’s center (measurement points P2/1 and P2/2). The HBM Spider Quantum X acquisition system was used to record the measurement data with 2 Hz frequency.

### 4.3. Test Results

The basic outcome of the push-out test is a load-relative slip curve. This plot (typically average of all slip measurements) enables us to determine the first slip load P_s_ and corresponding slip value δ_1_, both necessary for slip modulus k_slip_ estimation. Ultimate load P_u_ and corresponding ultimate slip δ_u_ let us to determine shear connecting capacity. The failure mode indicates the weakest elements of the connection. All of these outcomes are described and illustrated below for all specimens tested.

In specimen S1 a cracking sound was heard at a total load level of 255 kN. It was the first relative slip of 0.12 mm, when the adhesion between the GFRP and the concrete was lost (hereafter named debonding). As load continues, the second cracking sound and slip were noted at almost the same load level and finally two bolts on one side suddenly fractured at approximately 311 kN (Figure 7). Bolts on the other side of the specimen did not fail. Figure 8 shows the average load–slip curve of specimen S1. As far as the specimen’s failure mode is considered, the pure shear failure of both bolts were observed and neither concrete cracking nor bearing or shear-out failure of the laminate were exhibited (Figure 9).

In specimen S2, the loading phenomenon was very similar to that in S1, initial cracking sound was heard, and the first slip occurred at approximately 245 kN and the ultimate load was 315 kN. The bolts on one side suddenly fractured, while bolts on the other side did not fail. Concrete cracking and bearing or shear-out failure of the laminate were not observed on the respective surfaces of the specimen S2. However, two times bigger first slip value of 0.30 mm was observed in this case, due to premature slip of 0.065 mm that occurred at about 160 kN (Figure 10). This is probably attributed to the worse GFRP surface condition and induced that the bolts gradually slipped into a bearing region around the holes in the GFRP (Figure 11).

In contrary to the specimens S1 and S2, the S3 specimen exhibited almost no slip until the load of 405 kN was reached and the first significant slip took place. After that the specimen behaved almost linearly until the second debonding between the GFRP and LWC slab occurred at 458 kN. However, the careful checking of the specimen after failure revealed that the load applied to the specimen was transferred to the LWC slab mostly by the concrete surface protrusion (Figure 12). The protrusion was made during concrete casting due to false formwork execution. The slight initial slip at 140 kN occurred only on one side of the specimen due to initial crushing of protrusion and was negligible (Figure 13). When the protrusion had crushed at 458 kN, the load rapidly decreased to 281 kN and the specimen behavior came back to the expected one (as for specimens S1 and S2). The bolts on one side suddenly fractured at the ultimate load of 342 kN, while bolts on the other side did not fail. Concrete cracking and bearing or shear-out failure of the laminate were not observed on the respective surfaces of the specimen S3. The specimen S3 had higher ultimate shear capacity and smaller slip than S1 and S2, which is attributed to different performance in early load stage due to concrete surface protrusion.

The push-out experiment demonstrated that there were no significant differences between the behaviors of three specimens S1–S3 except concrete protrusion effect in specimen S3. The load–displacement curves of three specimens were almost parallel up to first slip (debonding), resulting in a similar stiffness of three specimens (Figure 14). (Note: the initial displacement in the range of 0–50 kN is due to steel package load adjustment). After two slips attributed to two slip planes of specimens, the load was transferred to the slabs solely by bolts. The behavior after debonding was always nonlinear up to the failure, attributed to yielding of bolts. Specimens finally failed due to pure shear fracture of bolts and no cracking, bearing or shear-out failure around the holes of the GFRP and concrete were observed. The results showed that the capacity of the bolts were sufficient to provide reverse strength after debonding and therefore the specimens did not fail immediately after debonding. The main quantitative outcomes of the push-out tests are provided in Table 3. As can be seen in this table the ultimate failure force was between 311–342 kN with a maximum difference of a 9.9%.

## 5. Fatigue Test

### 5.1. Test Setup

The main purpose of the fatigue test was to determine the fatigue life and a possible reduction of static strength of the shear connection subjected to unidirectional cyclic loading. The experimental program consisted of a series of 3 specimens F4–F6 tested with constant amplitude loading. The specimens manufactured for the fatigue test exhibited identical geometry to those of the static strength determination. The effect of fatigue loading were investigated with varying parameters: minimum load F_min_, peak load F_max_ and loading range ΔF. During the cyclic test, the load from the actuator load cell and the displacement of the specimen were measured, the latter by four LVDT’s in P1 and P3 measurement points. Cyclic tests were conducted under sinusoidal control waveforms with a load frequency of 2 Hz.

In three static tests performed previously (S1–S3) the mean value of the ultimate static load P_u,av_ was determined (Table 4). This value represented the reference parameter for the relative values of loading required for cyclic tests. Cyclic tests were conducted for loading range approximately 30%–60% of P_u,av_ with the stress ratio R = 0.1. Three load controlled cyclic tests on specimens F4, F5 and F6 were performed to determine the fatigue life N of the shear connection. The loading parameters of fatigue test are summarized in Table 5. After reaching the design number of cycles (2 million) one of these three test specimens did not fail and it was statically loaded to failure under displacement control to obtain the reduced static strength after high cycle preloading.

### 5.2. Test Results

During the cycling tests, the displacement range of the specimens was recorded while keeping the load range constant. The displacement was increasing rapidly due to crack initiation at the welding region as shown in Figure 15. Numbers of cycles at crack initiation and at failure are summarized in Table 6.

The general view of the fatigue failure mode of the specimen F4 is shown in Figure 7. Almost vertical displacement of one slab due to sudden fracture of two bolts was exhibited without affecting the second slab at all. To investigate failure mode, the concrete slab was separated from the GFRP tube and the fractured surfaces were examined. Fatigue failures occurred at weld toes (Figure 16). The examined fracture surfaces consisted of the typical dull fatigue fracture formed by propagating cracks and no bright forced fracture zones due to forced shear fracture were visible. The thorough examination revealed the crack initiation at point at the bolt shank in the heat affected zone above the melting line and successive horizontal crack propagation through the shank. In Figure 16 the GFRP laminate shear-out failure around the predrilled holes is also shown.

Since the specimen F6 did not fail after 2 million cycles of loading, the fatigue test was stopped and the specimen was loaded statically in the same way as specimens S1–S3 to evaluate possible reduction of the static strength of the shear connection subjected to fatigue loading. The specimen F6 exhibited no slip until the load of 320 kN was reached (Table 4). Similar to specimen S3 the load applied was transferred initially by the concrete surface protrusion. The protrusion was crushed at 365 kN, the load rapidly decreased, and the specimen started to transfer the load only by bolts. The bolts suddenly fractured at the ultimate load of 315 kN, what was similar ultimate value as for specimens S1 and S2. Failure mode was similar as well. Hence no reduction of the static strength of the shear connection was revealed when subjected to cyclic loading (Figure 17).

## 6. Discussion

### 6.1. Load-Slip Behavior

The load-slip curves based on push-out tests are evaluated to assess the shear capacity and the ductility of the novel shear connectors. The load-slip curve can be divided into three stages (Figure 8). Linear behavior of the load-slip curve was observed for the first stage up to the first slip. The second stage showed a sudden load reduction caused by loss of the adhesion (or debonding) between the GFRP and the concrete (however, no adhesive was applied). The linear increase and sudden decrease were exhibited the second time in specimens S1, S3 and F6 due to the second slab debonding. Nonlinear behavior of the load-slip curve was observed in the third stage, attributed to the hardening of the connector. In contrary to typical stud connection behavior the final stage showed no softening and no stiffness reduction after the load exceeded the ultimate resistance of the bolts. It can be seen from Figure 8 that the load-slip behavior of the GFRP-concrete specimens are more ductile than that of the typical steel concrete specimens. This may be a preferable failure mode for designing of FRP–concrete hybrid girders with the use of the novel shear connectors.

The load-slip behavior of the connector is an important parameter to calculate the hybrid girder overall deformation considering the slip effect [17] and to assess the composite action. The comparison of the slip modulus k_slip_ is often used to evaluate shear connections. The slip modulus is given as follows:k_slip_ = P_s_/n δ_1_(1)
where: k_slip_ is the slip modulus for a single bolt, n is the number of bolts in the specimen (here n = 4) and δ_1_ is the slip at the load of P_s_, after which the nonlinear behavior becomes notable.

The outcome of k_slip_ for the novel connector is listed in Table 4. For specimen 2 in parenthesis k_slip_ value is also given after deducting the premature slip 0.065 mm (Figure 10), which was rather untypical behavior of the specimen. The average slip modulus k_slip_ is approximately 523 kN/mm, indicating that the connectors can promote the composite action very efficiently, than other solutions, where k_slip_ was obtained in the range of –200 kN/mm [25,29,35].

### 6.2. Failure Mechanism

According to the findings published in the existing literature, there have been two typical failure modes in the push-out specimens [17,19,25,26]: bolt shank shear (mode 1) and FRP flange shear-out (mode 2). The first mode is similar as in steel–concrete connections in composite beams, the latter is commonly observed in FRP composite connections due to a relatively low shear strength of FRPs. Only first failure mode, bolt shank fracture, was found in the current tests, which revealed bolt shank fracture to be determined by cross-sectional area and material strength. These results match well with previous findings [19,25,26], allowing us to deduce that the use of steel studs in FRP and concrete connections is similar to their use in steel–concrete composite beams. The guidelines in the relevant codes (for example [32,33]) can be used to predict the strength of the novel shear connection.

Due to small modulus of elasticity of GFRP, the hybrid girder may have problems with serviceability limit states and especially in deflection. Therefore, the behavior of the connection should be rigid. Solution of connection with steel bolt shear connectors reached ductile behavior but the debonding between surface of GFRP and concrete slab occurred before the maximum load was reached. In order to increase the initial stiffness of the connection, coarse sand coating of the GFRP surface can be applied.

The weld toe fracture in welds joining shear bolts to steel plate was the crucial failure mode in fatigue. In Figure 16 the GFRP laminate shear-out failure around the predrilled holes is also shown. This failure mode exhibited in cyclic loading. Despite this mode was not decisive for the fatigue life of a specimen, an effective way to improve shear capacity is to add additional fiber layers (preferably 45°) to top flange’s laminate.

### 6.3. Shear Capacity

Experimentally obtained ultimate resistance of the shear connector per bolt (at failure load P_r_) was defined by Equation (2):P_r_ = P_u_/n(2)
where: P_u_—experimentally obtained maximum load of push-out tested specimens; n-total number of bolts in specimen (here n = 4).

The average ultimate resistance of the shear connector per bolt is 80.67 kN and minimum / maximum values are −3.6%/+6.0%, respectively (Table 4). The relevant values from the literature were typically in the range of 20–50 kN, depending on stud’s steel or bolt class and concrete class [17,19,25,26,35]. This difference indicates that the use of either welding or bonding improved the novel connection capacity. To predict the ultimate load of failure mode 1, the literature (e.g., [17,25,26]) and current design codes, including Eurocode 4 [32], have provided Equation (2) in Appendix A for similar failure in a steel–concrete composite beam. Because tensile strength could not be directly used to calculate shear failure, the reduction factor ψ (usually 0.6–0.8) was introduced to compute the shear strength of the connector; in Eurocode 4 ψ = 0.8 [32]. In the present study the average ψ was obtained as 0.9. However, test data from [17] and [26] suggest ψ = 0.6 for FRP–concrete shear bolt connections. The reason of this difference can be the combined use of bolts and epoxy adhesive in the novel connection, what may result in an increase in ultimate resistance of the bolts. The authors of [17] and [26] determined this reduction factor to be 0.6 for the bolted only specimens, not for those with the bolted—bonded connection. On the other hand, according to [25] the comparison of specimens with and without epoxy suggests that ψ is larger than 1.0 in the combined joints, which does not agree with current theory, but supports the abovementioned reason.

The ultimate strength of shear connection obtained from static tests was also compared with those calculated by design codes of Eurocode 4 [32]. In terms of bolt shank shear capacity, the average ultimate resistance obtained in the test is 11.8% higher (80.67/72.15 = 1.118) than the characteristic resistance of one headed stud according to Eurocode 4 [32]. Nevertheless, the code slightly underestimates the shear capacity of the connection, it can be used to predict the static strength of the connection and to check its ULS/SLS design provisions.

Taking into account test limitations it can be said that the design code estimates the shear strength of the novel connection quite properly, with acceptable accuracy and on safe side. When compared to maximum characteristic shear force per stud obtained in bridge design P_Ed_ = 21.98 kN (Table 1) the global safety factor of the shear connection can be estimated as follows: 80.67/21.98 = 3.67 showing the high safety, reliability and robustness of the novel connection system.

### 6.4. Fatigue Strength

Eurocode 4 [32] specifies the fatigue strength curve of an automatically welded headed stud as follows:(Δτ_R_)^m^·N_R_ = (Δτ_c_)^m^ × N_c_(3)
where: Δτ_R_ is the stress range; m is the slope of the fatigue strength curve with the value m equal to 8; N_R_ is the number of stress-range cycles; Δτ_c_ is the reference value at N_c_ = 2 million cycles with Δτ_c_ equal to 90 MPa.

For studs in lightweight concrete the fatigue strength should be determined in accordance with Equation (3) but with Δτ_R_ replaced by η_E_ Δτ_R_ and Δτ_c_ replaced by η_E_ Δτ_c_, where η_E_ is given in Eurocode 2 [34] as 0.955.

A comparison between test results and design values according to Eurocode 4 [32] is summarized in Table 6 and presented in Figure 18. The test results are in good agreement with the given prediction according to Eurocode 4 [32]. The fatigue life of the novel shear connection is slightly longer than the design value for typical shear studs, especially for higher stress range. From a linear regression analysis of the test results, the slope of the fatigue strength curve m and the reference value at two million cycles Δτ_c_ are calculated as 12.95 and 142.6 MPa, respectively. Thus, the fatigue strength obtained from the fatigue test is 142.6/(0.955· 90) = 1.66 higher when compared to the fatigue strength according to Eurocode 4 [32].

## 7. Conclusions

A reliable connection between FRP and concrete slab in hybrid bridge girders is important to prevent sudden and brittle failure. The use of steel shear connectors can reach this robustness. This study demonstrated the feasibility of the connection by shear bolt connectors between concrete slab and GFRP beam. Bolts together with adhesive achieved great properties in performed push out tests.

This study presents experimental results of push-out and fatigue tests on a novel shear connection system joining the GFRP laminated beam and the concrete slab of the hybrid bridge girder. Three static push-out tests and fatigue test were performed to evaluate advantages of steel shear connectors in promoting the connecting of GFRP girder and concrete slab. The load–slip curves, shear capacity, fatigue strength and failure mechanism of the novel shear connectors are discussed. The following conclusions can be drawn from this study:
•the load—slip behavior of the GFRP–concrete specimens is more ductile than that of typical steel-concrete specimens;•the average slip modulus k_slip_ value is approximately 523 kN/mm, indicating that the connectors can promote the composite action very efficiently;•bolt shank fracture is the only failure mode, found in the static tests, while the weld toe fracture in welds joining shear bolts to steel plate was the failure mode in fatigue; these may be a preferable failure modes for designing of FRP–concrete hybrid girders with the use of the novel shear connectors;•the average ultimate resistance obtained from the test is about 12% higher than the characteristic resistance of shear studs according to Eurocode 4 [32];•the fatigue strength curve slope m = 12.95 and the reference value Δτ_c_ = 42.6 MPa at two million cycles are determined; thus, the fatigue strength obtained from the test is 1.66 higher when compared to the fatigue strength of shear studs according to Eurocode 4 [32];•the global safety factor of the shear connection is estimated as 3.67 showing the high safety, reliability and robustness of the novel connection system,•Eurocode 4 [32] slightly underestimates the shear capacity and the fatigue strength of the novel connection; however, despite this conservatism this code can be used to predict the strength of the connection and to check its ULS/SLS design provisions.


The results of experimental research concerning the shear behavior of the novel connection system between the GFRP beam and the concrete slab revealed, that galvanized steel headed bolt connectors combined with epoxy adhesive may be a very good option to obtain ductile, safe, reliable and robust behavior for the shear connections of GFRP-LWC hybrid bridge girders subjected to static as well as fatigue loading. Therefore, it was decided to implement it as the shear connection in hybrid girders for the first Polish FRP bridge.

## 8. Patents

Patent No. 231211: “Road bridge superstructure”, E01D2/04, Polish Patent Office, Warsaw, Poland, 02/2019.

## Figures and Tables

**Figure 1 materials-13-02045-f001:**
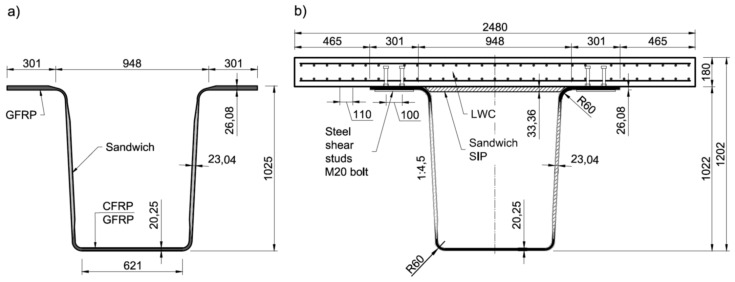
Bridge girder’s cross-section: (**a**) fiber-reinforced polymer (FRP) beam; (**b**) hybrid FRP- lightweight concrete (LWC) girder (unit: mm).

**Figure 2 materials-13-02045-f002:**
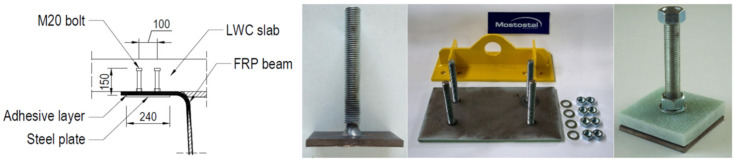
Idea and elements of novel shear connection system.

**Figure 3 materials-13-02045-f003:**
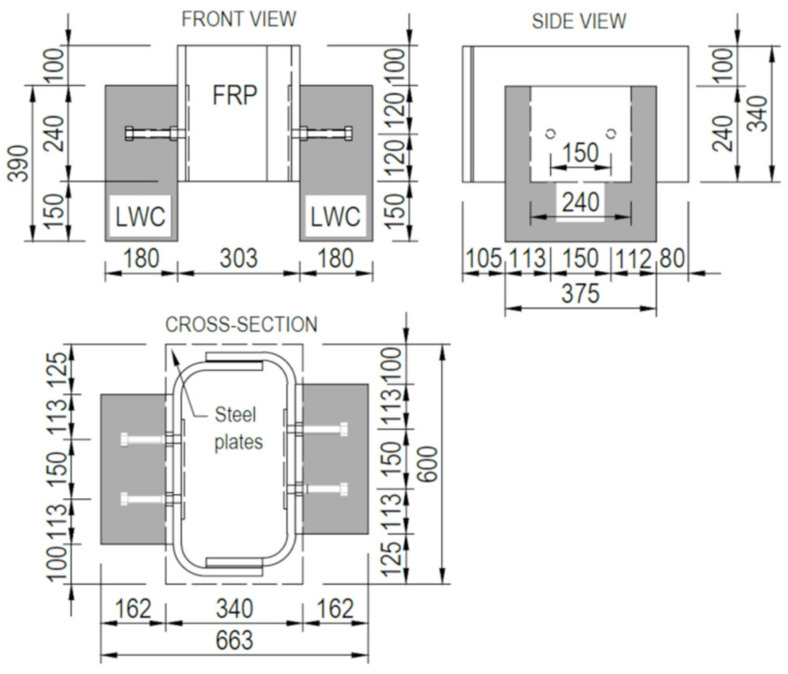
Geometry of push-out test specimen.

**Figure 4 materials-13-02045-f004:**
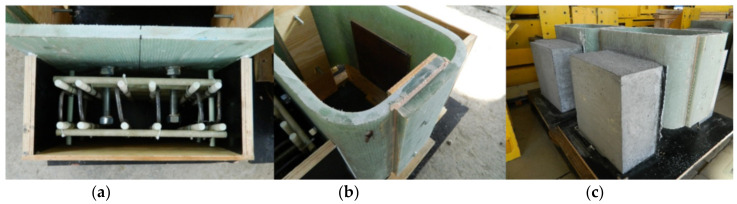
Specimen’s fabrication: (**a**) glass fiber-reinforced polymer (GFRP) reinforcement of concrete slab; (**b**) steel connectors inside the tube; (**c**) ready specimen.

**Figure 5 materials-13-02045-f005:**
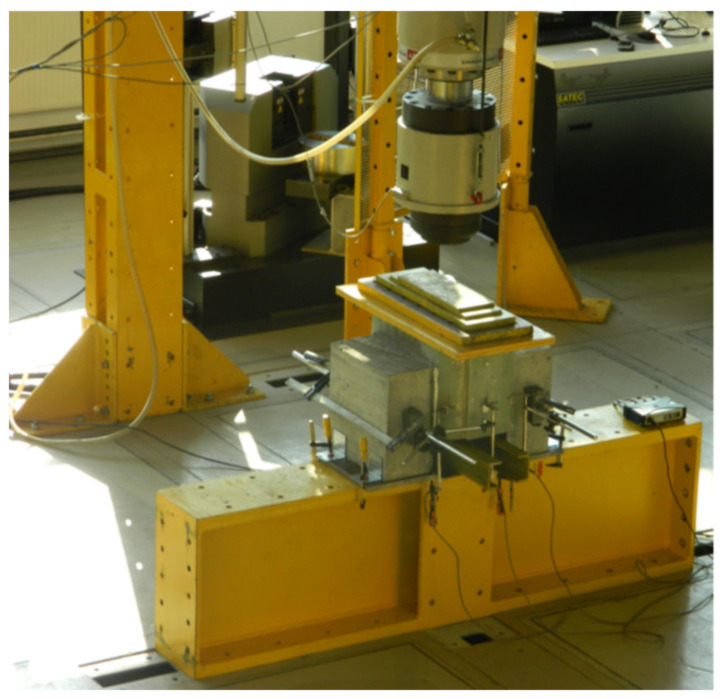
Push-out test set up.

**Figure 6 materials-13-02045-f006:**
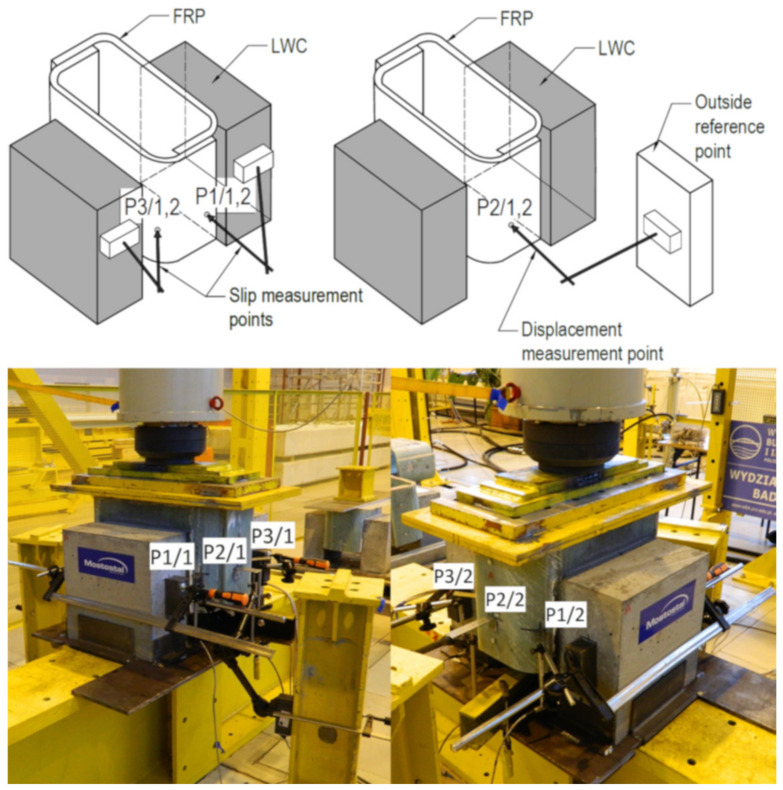
Location of linear variable differential transducer (LVDT) measurement points on specimen.

**Figure 7 materials-13-02045-f007:**
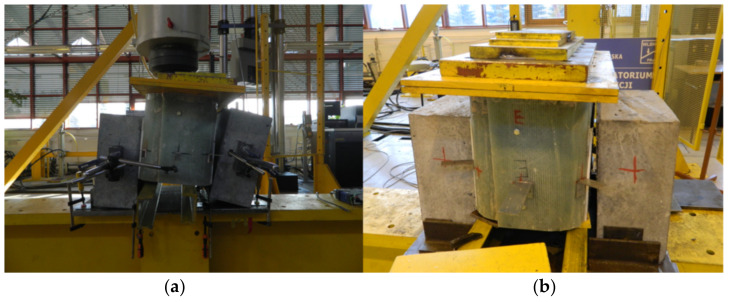
Failure modes: (**a**) static failure of specimen S1; (**b**) fatigue failure of specimen F4.

**Figure 8 materials-13-02045-f008:**
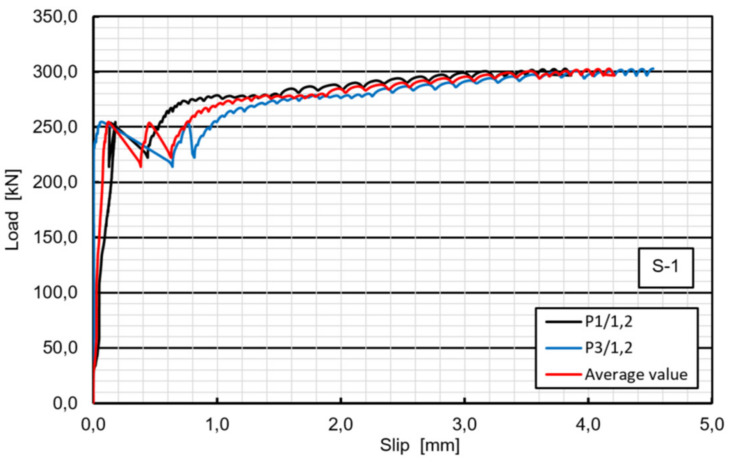
Load–relative slip curves of specimen S1.

**Figure 9 materials-13-02045-f009:**
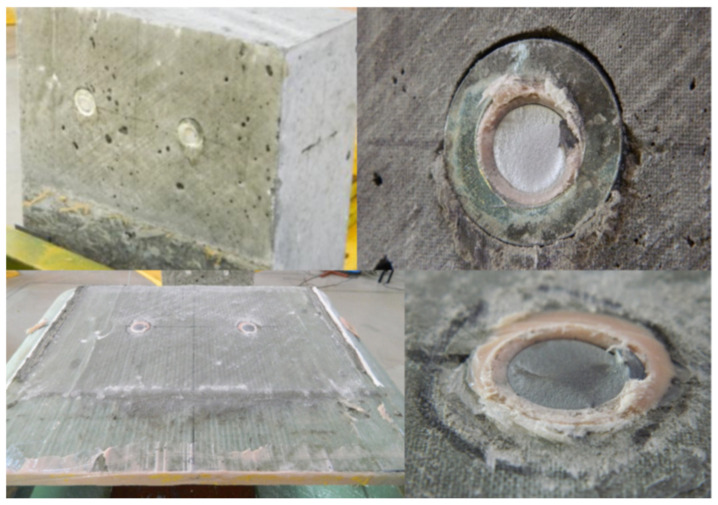
Failure mode of specimen S1: concrete surface—no cracks (**top**); GFRP surface—no bearing or shear-out failure (**bottom**).

**Figure 10 materials-13-02045-f010:**
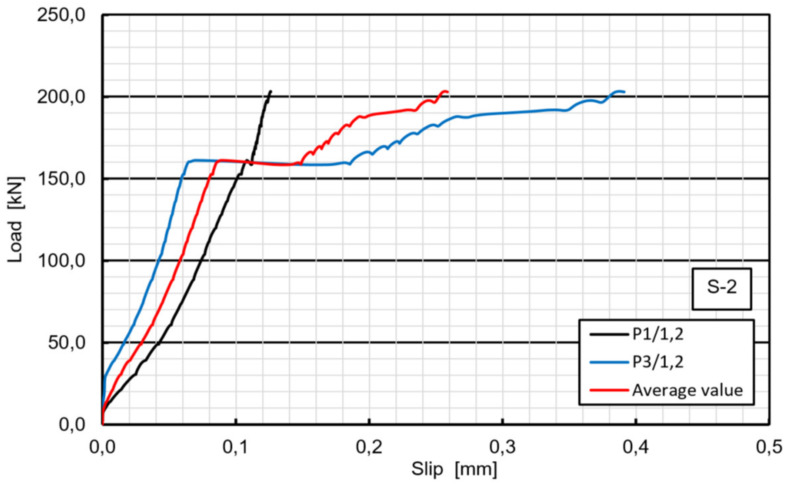
Load–slip curves of specimen S2 in the range of 0–200 kN: premature slip at 160 kN.

**Figure 11 materials-13-02045-f011:**
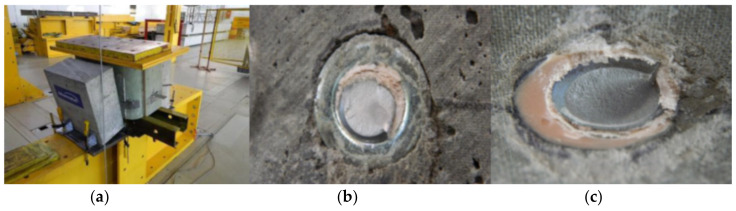
Failure mode of specimen S2: (**a**) general mode; (**b**) concrete surface—no cracks; (**c**) GFRP surface—initial bearing (right).

**Figure 12 materials-13-02045-f012:**
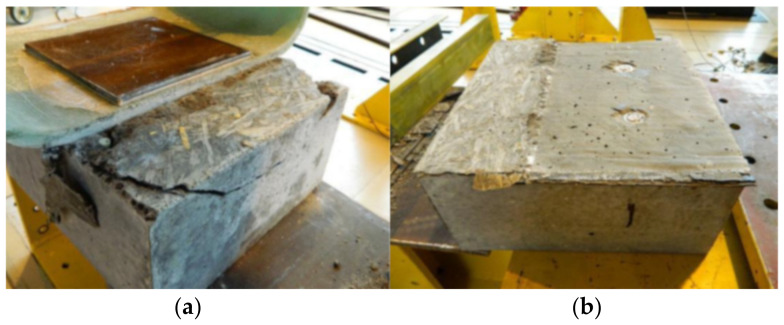
Failure mode of specimen S3: (**a**) concrete surface protrusion crushed; (**b**) concrete surface—no cracks.

**Figure 13 materials-13-02045-f013:**
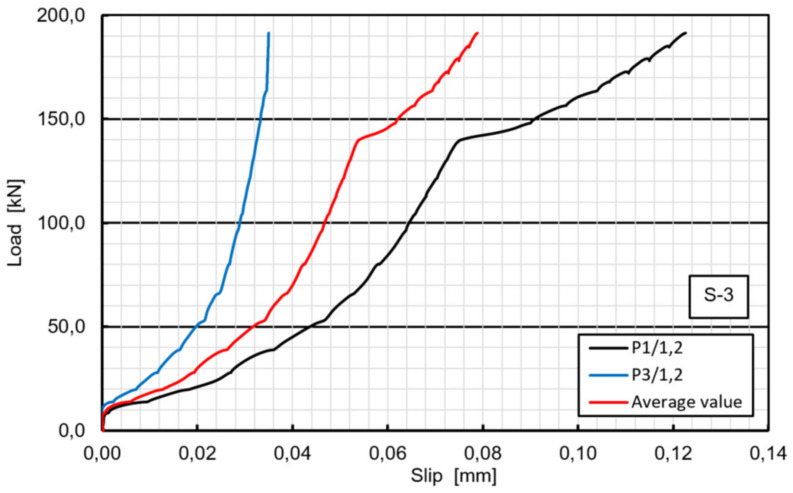
Load–slip curves of specimen S3 in the range of 0–200 kN: premature slip at 140 kN (one side only).

**Figure 14 materials-13-02045-f014:**
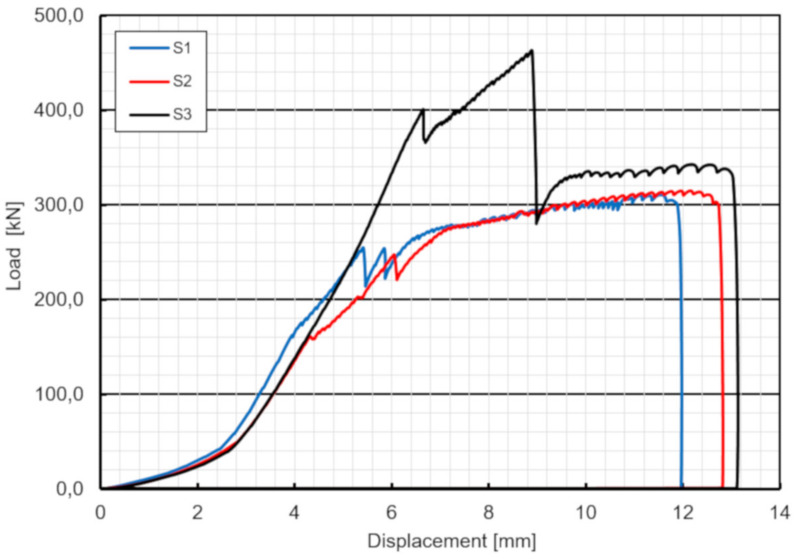
Load–displacement curves of three specimens.

**Figure 15 materials-13-02045-f015:**
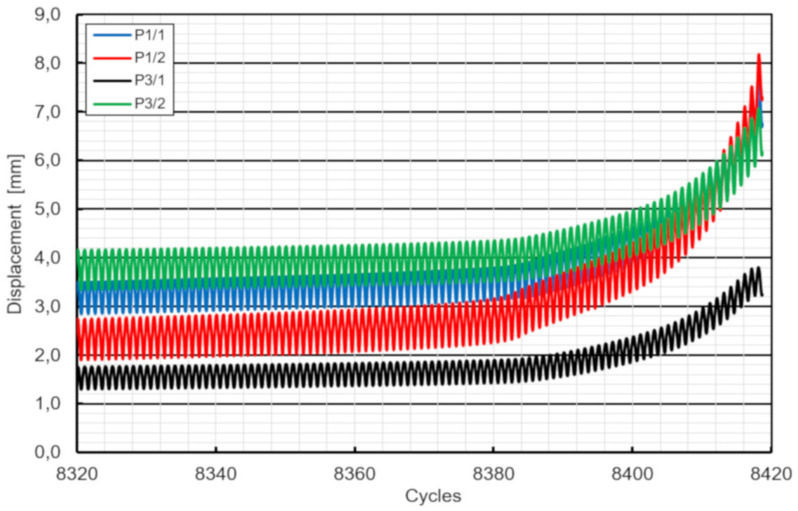
Load–displacement behavior of specimen F4 under cyclic loading (final range).

**Figure 16 materials-13-02045-f016:**
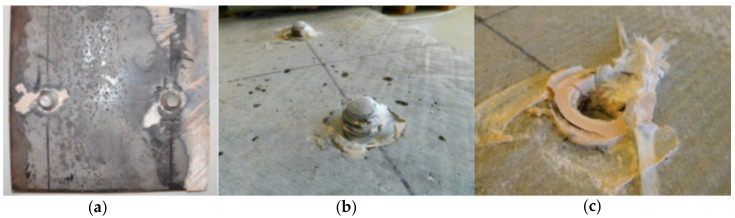
Fatigue failure mode of specimen F4: (**a**) fracture at weld toe on steel plate; (**b**) fractured bolt shanks; (**c**) slight shear-out failure in GFRP laminate.

**Figure 17 materials-13-02045-f017:**
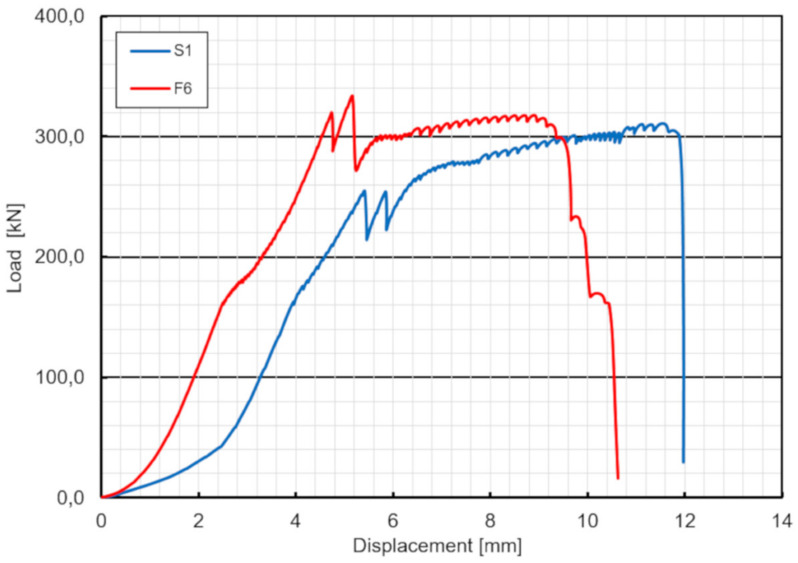
Load–displacement curves of specimens S1 and F6.

**Figure 18 materials-13-02045-f018:**
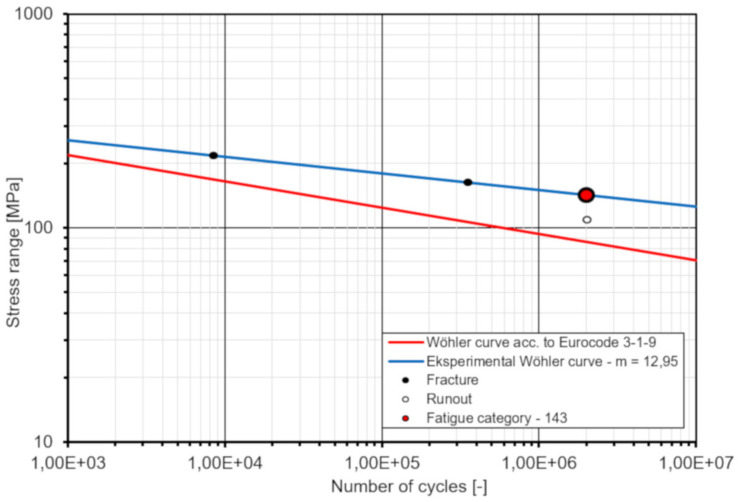
Standard and experimental S–N curves for shear studs (bolts).

**Table 1 materials-13-02045-t001:** Shear forces according to bridge design.

Load	Total Shear Force Per Unit Length v_L,Ed_	Maximum Shear Force V_Ed_	Maximum Shear Force Per Study P_Ed_
	[kN/m]	[kN]	[kN]
Characteristic	488.40	349.44	21.98
Design SLS	488.40	349.44	21.98
Design ULS	696.43	523.02	31.34

**Table 2 materials-13-02045-t002:** Mechanical properties of laminas used in specimen fabrication.

Constant, Parameter	Unit	Symbol, Direction		Lamina	
			X-E ±45°	B-E 0/90°	U-E 0° (90°)
			1210 g/m^2^	800 g/m^2^	1210 g/m^2^
Longitudinal modulus of elasticity	GPa	E_x_	20.50	20.00	42.13
		E_y_	20.50	20.00	10.87
Transverse modulus of elasticity	GPa	G_xy_	3.90	3.90	4.40
		G_yz_	3.04	2.83	2.71
		G_xz_	3.04	2.83	2.71
Poisson’s ratio	-	ν_xy_	0.019	0.029	0.29
		ν_yz_	0.019	0.029	0.075
		ν_xz_	0.019	0.029	0.075
Tensile strength	MPa	X_t_	520.0	522.0	855.0
		Y_t_	520.0	522.0	44.0
Compressive strength	MPa	X_c_	320.0	321.0	537.0
		Y_c_	320.0	321.0	84.0
Shear strength	MPa	S_xy_	60.0	60.0	51.0
		S_yz_	30.0	30.0	25.0
		S_xz_	30.0	30.0	25.0

**Table 3 materials-13-02045-t003:** Mechanical properties and density of LWC 35/38 class.

Constant, Parameter	Unit	Symbol, Direction	Value
Modulus of elasticity	GPa	E_c_	26.986
Tensile strength	MPa	f_t_	2.46
Compressive strength	MPa	f_c_	25.76
Ultimate compressive strain	[‰]	ε_lcu1_	1.749
Density	kg/m^3^	ρ	1968

**Table 4 materials-13-02045-t004:** Outcomes of push-out tests.

Specimen	First Slip			Failure			
	Slip Load P_s_	First Slip δ_1_	Modulus k_slip_	Ultimate Load P_u_	Ultimate Slip δ_u_	Ultimate Resistance P_r_	Mode
	[kN]	[mm]	[kN/mm]	[kN]	[mm]	[kN]	
S1	255	0.12	531.3	311	4.2	77.75	bolt shank fracture
S2	245	0.30 (0.235)	204.2 (260.6)	315	4.3	78.75	bolt shank fracture
S3	405	0.16	632.8	342	4.5	85.50	bolt shank fracture
F6	320	0.12	666.7	315	4.9	78.75	bolt shank fracture
Average ^1^			522.8	322.67		80.67	

^1^ Without specimen F6.

**Table 5 materials-13-02045-t005:** Fatigue test parameters.

Specimen	F_min_	F_max_	R = F_min_/F_max_	ΔF	ΔF/P_u,av_
	[kN]	[kN]		[kN]	[%]
F4	22.2	222.2	0.1	200	61.9
F5	16.6	166.6	0.1	150	46.5
F6	11.0	111.0	0.1	100	31.0

**Table 6 materials-13-02045-t006:** Fatigue test results.

Specimen	Stress Range Δτ_R_	Crack Initiation	Number of Cycles at Failure N_R_	Log (N_R_)/Log (N_c_) acc. to [32]
[MPa]	[cycles]	[cycles]
F4	217.7	8.384 × 10^3^	8.412 × 10^3^	
F5	163.3	2.945 × 10^5^	3.488 × 10^5^	1.66
F6	108.9	–	run-out

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
