# Peer review of "Experimental Study on a Novel Shear Connection System for FRP-Concrete Hybrid Bridge Girder"

_materials, 2020, doi:10.3390/ma13092045_

Round 1
Reviewer 1 Report
The paper is interesting and well-written. The quality of presentation is very good. I think the only minor problem of the paper is the literarure review and the comparison between present results and the exisitng literatures.
Author Response
The authors thank very much the Reviewer for valuable remarks, which have improved the text considerably. Please find attached our answers and comments to the remarks.

Reviewer 2 Report
The paper deals with materials and structural tests performed to support the designing process of the first FRP bridge built in Polish in 2015. In particular, reference is made to the shear behavior of the novel connection system between the FRP beam and the concrete slab. The paper summarizes the results of the static and fatigue tests and discusses the structural performance of the proposed connection system. The main question is related to the applicability of the EC4 for the design of the proposed connection system. As a matter of fact, in the discussion it is shown that the EC4 provide a reasonable estimation of the shear strength, but in several other sections it is stated that EC4 does not. The same hold for the fatigue strength. The author should first clarify this point. The following additional question are raised to the authors’ attention:
- Please, specify the unit in Fig. 1 is mm and verify if “M20 belt” is correct or not;
- The sandwich SIP panel listed in Fig. 1 is not commented in the text;
- In Fig. 1, 4 bolts are shown but the text mention at line 108 a total of 8 bolts is mentioned. Why?
- Details of the shear connection design (section 3) should be moved in a specific appendix since they are not the main purpose of the paper;
- In Table 1, the third column is probably the shear force for stud since it is obtained by dividing the first column by the number of stud per unit length. Please check;
- During the static tests, the rotation of the two lateral LWC blocks was prevented or not? Please comment;
- The slip was measured on two sides of the specimens but just one graph is reported in Figs. 8 and 10? Why;
- Please, replace “P32” with “P3/2” at line 268;
- Why the load-slip graph for specimen S3 is not reported?
At line 409-410 it is stated that no softening was observed but the test was performed in load control. Please explain;
Author Response

(The authors gave the same response as above.)

Reviewer 3 Report
It is rather a test report on shear connection system than a scientific research on materials design and properties characterization. However, the presented experimental work is very attractive, while the test program was well described. Steel shear connectors were designed to prevent failure between concrete slab and bridge girders. Here are some comments for improvement:
1, please revise your manuscript with more detailed materials characteristics in your experimental test, which will let your manuscript fulfill the scopes of Materials.
2, it might be better to put the failure modes in one figure. In this case, the differences between failure modes can be presented clearer.
Author Response

(The authors gave the same response as above.)

Round 2
Reviewer 3 Report
I am satisfied with the revision.